# Regional Travel as an Alternative Form of Tourism during the COVID-19 Pandemic: Impacts of a Low-Risk Perception and Perceived Benefits

**DOI:** 10.3390/ijerph18179422

**Published:** 2021-09-06

**Authors:** Xin Wang, Ivan Ka Wai Lai, Quan Zhou, Yu He Pang

**Affiliations:** Faculty of International Tourism and Management, City University of Macau, Macau 999078, China; xwang@cityu.mo (X.W.); ivanlai@cityu.mo (I.K.W.L.); zhouquan1997@foxmail.com (Q.Z.)

**Keywords:** low-risk perception, perceived benefit, alternative tourism, Greater Bay Area in China, travel during and after the COVID-19 era

## Abstract

Previous COVID-19 tourism research has not considered the positive impact of a low-risk perception and a perception of the benefits of regional travel on taking alternative tourism. This study attempts to fill the research gap and examine the positive effect of these perceptions on tourists’ attitudes to regional travel and intentions to undertake regional travel during the COVID-19 pandemic. A survey of 278 respondents confirmed that the perceived benefit positively influences tourists’ attitudes and travel intentions, but that a low-risk perception only positively affects their attitudes. This study contributes to tourism risk management research by introducing the concept of a low-risk perception as a positive factor. For tourism recovery, it finds that relaxation, value, and convenience are benefits to drive people to travel.

## 1. Introduction

The current outbreak of the novel coronavirus SARS-CoV-2, officially named COVID-19, has classified as a major public health emergency, and spread to many countries [1]. With the growing rates of case notifications, the World Health Organization (WHO) Emergency Committee declared a global health emergency on 30 January 2020 [2]. This has had a strong impact on all areas of life, especially the tourism industry, across the world. Many world-leading experts believe that COVID-19 could become a seasonal or an endemic virus, such as flu or the common cold, with infections ebbing and flowing throughout the years to come [3,4,5]. During the COVID-19 pandemic, people have avoided using public transport, where there is a high risk of virus transmission, and have chosen personal modes of transport. With the changes in travel patterns, the frequency of non-mandatory trips (for shopping, leisure, social reasons, etc.) has been reduced and the length of such trips has shortened [6]. Given the relatively long period during which COVID-19 has continued, the UNWTO (The World Tourism Organization) has published its Global Guidelines to Restart Tourism, to guide the safe restarting of travel [7]. In responding to the restarting of tourism, researchers have argued that there will be some new travel patterns, such as regional tourism, health tourism, and rural tourism. Previous COVID-19 tourism research has not considered the positive impact of a low-risk perception and a perception of the benefits of regional travel on taking alternative tourism [8-9].

The pandemic has brought the world’s travel traffic to a standstill, and many countries have banned foreigners from entering the country and closed their foreign borders. The degree of rigor (or looseness) of COVID-19 prevention and control enables us to set up three categories: China’s national strictly precise prevention and control model; Europe’s flexible prevention and control model; and America’s free and loose prevention and control model. Restrictions on freedom of movement, closure of borders, and people’s fear of infection have completely caused people’s anxiety and greatly impacted the tourism industry [10]. The Asmundson’s research shows that anxiety, a state of tension and worry, mainly affects people’s behavioral intentions. In this pandemic, the tension and anxiety caused by policies such as restricting free movement have even made people feel fear [11]. On the whole, it is affecting public physical and mental health.

Traveling can relieve psychological pressure and regulate anxiety. During the recovery of the tourism market, the attraction radius of tourist destinations has shrunk, the proportion of local tourists has increased, and long-distance tourists have dropped significantly [12]. After the initial wave of the pandemic, in March 2020, China implemented inter-province controls under which residents of China who have undergone a nucleic acid test and received a green health code after seven days can move freely around their province [13]. People in China can therefore undertake ‘regional travel’. If this succeeds, regional tourism in a small area will become a new travel pattern for cities all over the world as an alternative form. It is therefore necessary to study the factors that influence whether people undertake regional travel.

In the study of tourism destinations, the decision of tourists to travel can be divided into two aspects: perceived risk [14] and perceived benefit [15]. Researchers have found that people’s perception of risk influences their choice to travel during the COVID-19 pandemic [16]. The risk perception arising from the epidemic can be divided into cognitive risk perception and affective risk perception [17]. Both cognitive and affective risk perceptions exert a negative influence on behavioral intention [14]. Under the bad apple principle, if there is a box of rotten apples, people tend to select the one that is not so rotten [18]. During the COVID-19, the travel craving, which also enables people to choose a good physical and mental tourism activities in low-risk areas. These cognitive emotional activities haves also played a certain promotion of the recovery of tourism [19]. If this applies, when a tourist considers two risky tourist destinations, he/she will select the low-risk one. Compared with normal travel, regional travel is a low-risk travel activity, and people may therefore have an attitude and behavioral intention in favor of this low-risk travel pattern. The perception that regional tourism has a low-risk may then affect people’s attitude and behavioral intention towards regional travel during and after the COVID-19 pandemic. Although risk perception and risk management research has been widely studied, there has been no study of the concept of a low-risk perception of an alternative type of travel in a positive perspective.

Other than a low-risk perception, people may also consider the benefits of undertaking regional travel during and after the COVID-19 pandemic. The perceived benefit of travel refers to the desirable gains sought by tourists from travel experiences [15]. Although researchers have indicated different benefits of undertaking (normal) travel, the benefits of undertaking regional travel may vary. In the current situation, tourists may consider value benefit, convenience benefit, and relaxation benefit [20,21,22]. However, among studies of post-COVID-19 tourism recovery, especially in alternative tourism, none has verified the effect of perceived benefit on tourists’ attitude and intention towards alternative tourism, including regional travel. Knowing the beneficial motivations helps to promote regional travel as well as alternative tourism, and to accelerate the post-COVID-19 tourism recovery. Therefore, it is essential to study the effect of perceived benefit and a low-risk perception on tourists’ attitudes and intentions towards regional travel. 

The aim of this study is to investigate the positive effect of a low-risk perception and the perceived benefit of undertaking regional travel on tourists’ attitudes and intentions towards regional travel. There are three research contributions. First, this study introduces the concept of a low-risk perception, which may positively influence tourists’ attitudes and their choice of a tourist destination. Therefore, researchers could consider including this concept (low-risk perception) in their research models when studying the phenomenon and problems of tourists’ choice of tourism destinations. Secondly, although researchers are studying ways in which tourism will recover after COVID-19, they have not investigated the effect of perceived benefit, despite this being a core motivation for travel. Therefore, this study fills this gap by evaluating the role of convenience benefit, value benefit, and relaxation benefit in promoting the recovery of the tourism industry. Thirdly, this study contributes to tourism risk management research by understanding the effects of low cognitive risk perception and low affective risk perception. Using the results of the study, practical recommendations are provided to governments and tourism enterprises for the development of tourism recovery strategies during and after the COVID-19 pandemic, so that the tourism industry can gradually recover.

## 2. Literature Review and Development of Hypotheses

### 2.1. COVID-19 Tourism Research

COVID-19, a sudden public health incident, has changed people’s travel patterns. Bhagat et al. [23] found that people in India are traveling less during the COVID-19 outbreak. They prefer walking and cycling to using public transport. In addition, Abdullah et al. [24] found that people in various countries around the world reduced the number of journeys they made and kept their travel distances short during the COVID-19 outbreak. People used safer (in terms of infection) modes of transport while they were traveling. Furthermore, those factors that generally influence transport choice, such as saving travel time, comfort, and cost, became less important during the COVID-19 pandemic. These changes in travel patterns provide an opportunity for sustainable and proximity tourism [25]. Within developed countries and emerging economies, where the majority of world travel requirements are centered, up-close travel holds promise to help to save the sector [26]. Samarathunga and Gamage [27] argued that, in the post-COVID-19 tourism renaissance, alternative tourism will emerge to replace the mass tourism phase with its high upside potential, and they thus identify emerging tourism products in niche tourism concepts such as health and wellness tourism, Ayurveda and spiritual tourism, rural tourism, agri-tourism, and ecotourism. Among the small-scale community regional control of the tourism industry, community security affects public risk perception and further affect health tourism intention [28].

In studying travelers’ attitudes to alternative travel during the COVID-19 pandemic, researchers have found that risk perceptions have a significant impact on individual decision-making. For example, fears deter people from traveling (except for non-contact travel) during a pandemic [14]. Furthermore, researchers have only focused on the health risk concerns associated with COVID-19 travel; few studies have been carried out on the benefits of alternative tourism. Recently, Nordin and Jamal [29] summarized the benefits of hiking tourism, which include improvements in quality of life and satisfaction, reductions in anxiety, and increased overall well-being of visitors. Wen et al. [30] suggested that, after periods of overwhelming stress and outbreaks, the benefits of nature-based activities (hiking, trekking, wildlife observation, or nature interpretation) include being able to breathe fresh air and becoming connected with nature and rejuvenation. Since different travel patterns provide different kinds of benefits, no study has investigated the benefits of undertaking regional travel. In addition, no study has investigated how the perceived benefit affects tourists’ attitudes and intentions towards alternative tourism.

### 2.2. Risk Perception

#### 2.2.1. Travel Risks

Risks refer to events associated with unanticipated and undesirable outcomes [31]. Based on previous literature from tourism scholars, Park and Reisinger [32] categorized risks during international travel into two types: natural disasters and travel risks. Volcanic eruptions, tornadoes, floods, tsunamis, droughts, hurricanes, heat waves, cyclones, earthquakes, avalanches, typhoons, landslides, winter storms, and wildfires are natural disasters [33]. By contrast, travel risks are mainly of 13 different types: cultural, crime, equipment/functional, financial, health, natural disaster, physical, political, psychological, satisfaction, social, terrorism, and time risks [34]. 

Previous studies have found that tourism is vulnerable to external environmental factors like natural and man-made disasters [35]. The risks associated with human activities even affect people’s willingness to travel [36]. For example, volcanoes, tsunamis, earthquakes, and typhoons, among the natural disasters that may occur in tourist destinations [37,38,39,40,41], and crime risk, political risk, and terrorism risk, among the man-made tourism risks, can all affect tourists’ scheduled travel plans [42,43,44,45,46,47]. During the COVID-19 period, most of these risks have not been applicable, and people have mainly considered the health risks. Health risks in tourism are associated with potential health hazards during travel to a place [48]. Since previous studies in tourism risk management were of little help in researching travel during the COVID-19 pandemic, many researchers have recently studied the risks of COVID-19 in undertaking normal travel. From another angle, low-risk can be used as a positive factor in the public health field to alternative tourism in the post-COVID-19 era.

#### 2.2.2. High-Risk Perception Arising from COVID-19

With the outbreak of COVID-19, travelers’ behavior and choice of destination have been compromised by worries about the perceived risks to their own health from contracting this infectious disease [49]. Perceived health risk refers to the reaction of tourists and hotel patrons when they feel their health is threatened by uncontrollable factors or events, such as a pandemic [50]. During the COVID-19 pandemic, people did not consider booking cruises as a leisure activity, and the perceived health risk was an important factor: It had a clear and strong negative impact on behavioral intentions [16]. The COVID-19 pandemic has led to a high level of perceived health risk for tourists when visiting destinations or hotel facilities. Even after the COVID-19 pandemic, researchers predicted that the majority of tourists will be unwilling to travel because of health concerns, making it essential for hotel and tourism practitioners to implement risk-reduction strategies [50]. 

The perception of risk has been studied by researchers who have looked at both the cognitive and the affective dimensions [17]. While cognitive risk perception consists of an individual’s perception of the susceptibility and seriousness of risk, affective risk perception relates to the individual’s personal anxiety or concern regarding the risk they face [51]. Both cognitive risk perception and affective risk perception have significant negative effects on behavioral intentions [14]. There have been studies on the influence of perceived health risk on international travel in the aftermath of the pandemic; travelers perceive the possibility of exposure to a health risk, so their perception of risk is increased, thereby increasing their mental wellbeing and perceived uncertainty [52]. Such a health risk, in turn, generally contributes to uncertainty in tourists’ decisions to travel abroad [53].

#### 2.2.3. Low-Risk Perception for Regional Travel

When the destination is accompanied by a degree of risk that tourists’ personal safety may be endangered, implying unpleasant consequences, tourists perceive the place as risky or unsafe, and they may then refuse to travel [54]. Therefore, a high-risk perception negatively affects tourists’ behavioral intentions [14]. As a result, instead of going to high-risk areas, travelers choose areas with a lower level of risk. They try to select a place where the risk is low. All current studies on tourism risk consider a high-risk perception as a negative factor that negatively affects tourists’ behavioral intentions. However, no research has explored the effect of a low-risk perception on the choice of tourism destination.

The cities in the Guangdong–Hong Kong–Macao Greater Bay Area (GBA) in China (excluding Hong Kong) are low-risk areas. As one of the most open and economically dynamic regions, the GBA has obvious advantages in terms of location, not only in the coastal region of China, but also in the ‘Belt and Road’ construction. The region is also home to many tourist attractions, such as the world’s largest marine theme park, Zhuhai ChangLong Ocean Park, and the world’s culinary capital, Foshan Shunde. Not only is the region rich in tourism resources, but the current COVID-19 situation is well-controlled, with both Guangdong Province and the Macao SAR falling into a low-risk area for COVID-19. There are, therefore, fewer travel restrictions for Macao residents, and travel to the GBA (excluding Hong Kong) is feasible and safe under current conditions.

In studies of risk perception, a meaningful mediator between risk perception and behavioral intentions is attitude [14]. People’s risk perception may, more or less, correspond to the actual situation, but since it implies an expectation of loss, it is highly likely to affect their attitude towards a behavior [55]. However, a lower level of risk may lead to a positive attitude. Guangdong province is currently a low-risk area. The Guangdong provincial government has put forward requirements for the prevention and control of COVID-19, and the COVID-19 position remains stable overall. Travel within a region is a tourism pattern that is safe for local tourists, so people in the GBA may have an intention to undertake GBA travel. Therefore, the following two hypotheses are proposed:

**Hypothesis** **1 (H1).**
*Tourists’ low-risk perception for undertaking GBA travel has a positive impact on their attitude towards GBA travel.*


**Hypothesis** **2 (H2).**
*Tourists’ low-risk perception for undertaking GBA travel has a positive impact on their intention to undertake GBA travel.*


### 2.3. Perceived Benefit

Perceived benefit is defined as a consumer’s conviction of the degree to which they will be better off after buying and/or using an item [56]. Those who perceive a greater benefit from a particular behavior will be more likely to carry out that behavior [57]. 

Perceived benefit varies depending on the consumption environment and the context in which this consumption takes place, as it is a key part of consumer choice [21]. Based on a survey of street food, Choi et al. [21] argued that street food has two perceived benefits that affect consumers’ behavioral intention: value and convenience. For travel, researchers have identified different types of travel benefits. For example, relaxation, excitement, social, and exploration benefits [58]; experiential, relaxation, and health benefits [15]; and relaxation, health, and experience benefits [22]. For different audiences and situations, travelers can derive four types of benefits from travel agents: financial benefit, emotional benefit, expertise, and support [59], while medical tourism can give tourists benefits from the quality of medical treatment, the waiting time, and the cost of medical treatment [60]. Using different research settings, researchers have considered different dimensions for the perceived benefit of going to different tourist destinations.

Currently, only tourist motivation in the COVID-19 situation has been studied by relevant scholars. No studies have identified the benefits of alternative tourism. In China, the COVID-19 pandemic is under control, and the social and economic recovery phase has begun [61]. Many tourist attractions in the GBA offer free admission or discounts to attract tourists. Restaurants, hotels and accommodation, entertainment venues and shops have all adopted actions to promote consumption by tourists. Furthermore, people in the GBA can obtain travel benefits, such as convenience benefits and relaxation benefits, when taking a short trip within the GBA. Therefore, people in the GBA may consider undertaking GBA travel to obtain certain types of travel benefits. If so, there may be a positive relationship between perceived travel benefit and intention to undertake GBA travel. On the basis of the above, the following research hypotheses are proposed:

**Hypothesis** **3 (H3).**
*The perceived benefit of undertaking GBA travel has a positive impact on tourists’ attitude towards GBA travel.*


**Hypothesis** **4 (H4).**
*The perceived benefit of undertaking GBA travel has a positive impact on tourists’ intention to undertake GBA travel.*


### 2.4. Travel Attitude and Behaviour for Low-Risk Tourism

Attitude is widely defined as an affective evaluation of an object or behavior [62]. With reference to the theory of planned behavior, attitude is strongly linked to tourists’ behavioral intention [63]. Lam and Hsu [64] found that tourists’ attitude is a determinant of their behavioral intention towards a destination. Therefore, tourists with a positive attitude towards GBA travel may be more willing to undertake GBA travel. The following hypothesis is consequently proposed:

**Hypothesis** **5** **(H5).**Attitude towards GBA travel has a positive impact on tourists’ intention to undertake GBA travel.

Based on the above five hypothesis, the research model of this study is shown in Figure 1.

## 3. Methodology

### 3.1. Research Setting

Macao is situated in the Pearl River Delta on the south-eastern coast of China, at longitude 113°35′ E and latitude 22°14′ N, about 60 kilometers east-north-east of Hong Kong. Macao has a total area of 32.9 square kilometers and a population of approximately 682,800. It is an international free port, a world center for tourism and leisure, and one of the four major gambling cities in the world. Macao is a part of the GBA. Since 2007, China’s high-speed railway network has been rapidly growing, and it now covers almost every city in the GBA. Improvements in regional transport accessibility further facilitate regional tourism within the GBA. Because of its small size and limited leisure resources, Macao’s residents like to visit GBA cities in their short holidays. 

With improved medical care and better prevention and control methods, the outbreak in Macao is well under control. By 15 March 2021, the Novel Coronavirus Infection Response Coordination Centre had reported no local cases (including those without symptoms) for 351 consecutive days and no new cases for 37 consecutive days; vaccination against the new coronavirus is steadily underway. The above-mentioned general environment provides the basic conditions allowing Macao residents, who are in a low-risk area for COVID-19, to travel outside. Since May 2020, Macao residents have been able to travel to cities in the GBA after having their nucleic acid tested and holding a green health code. The infrastructure and this favorable policy support Macao’s residents in traveling to GBA cities. During the COVID-19 pandemic, traveling to other places has become an ardent hope for Macao’s residents. Therefore, the choice of Macao as a study site during COVID-19 can demonstrate the impact of residents’ perceptions of the risks and benefits of regional tourism. In addition, there are no new locally transmitted COVID-19 cases, so it is safe for interviewers to conduct a face-to-face systematic survey in Macao.

### 3.2. Measurement Scales and Questionnaire Design

A questionnaire survey was used in this study to test the proposed hypotheses. For the low-risk perception, the measurable items of ‘low cognitive risk perception’ (likelihood … in general, likelihood … compared to other cities, likelihood … compared to other diseases, likelihood of dying) and ‘low affective risk perception’ (worried … myself, worried … family members, worried … in my region/occurring, and worried … a health issue) were revised from Bae and Chang [14] to alter the meaning so that it measures a low-risk perception. For example, the original item, ‘There is a high likelihood of acquiring COVID-19 in general’, was revised to ‘There is a low likelihood of acquiring COVID-19 when I travel to cities in the GBA in general’. The measurable items for attitude (important, useful, pleasant, and interesting) were derived from Wang et al. [22]. An example of these items is ‘I think current travel to cities in the Greater Bay Area is pleasant’. The measurable items of travel intention are borrowed from Sánchez et al. [65]; an example is ‘I intend to travel to cities in the Greater Bay Area as soon as I can’.

Choi et al. [21] showed that the two dimensions of perceived benefit for street food are related to ‘economy’ (large serving size, affordable price, reasonable food prices, and food value for money) and ‘convenience’ (eating convenience, easy accessibility, and prompt service). Under the influence of COVID-19, companies in the global tourism industry have been shutting down one after another. Tourism companies in the GBA are also facing great pressure to survive. In order to respond actively to the impact of COVID-19, the Guangdong provincial government has, first, increased its financial support to stimulate people’s consumption to drive economic recovery, while various companies in the tourism industry have actively engaged in self-help through offering discounts and issuing consumer vouchers. Secondly, the government has carried out reform to facilitate entry and exit to the GBA, so that tourists can still feel the economic benefits and convenience benefits of traveling in the GBA during the COVID-19 period. This study therefore chooses Choi et al.’s [21] scale to measure the perceived value benefit and perceived convenience benefit of undertaking GBA travel. 

The literature discussed above has shown that relaxation is regarded as a common benefit of tourism [15,58,66]. COVID-19 has made people more likely to have psychological problems such as anxiety and tension because of living in a fixed and closed environment. People will choose to have a relaxing trip, under the premise of relative safety. Therefore, this study also includes the items on perceived relaxation benefit (renew my energy, get away from tedious daily life, relieve tension or stress, refresh myself, and make me relax) from Wang’s [22] study as the third dimension of the perceived benefit.

This study employs a 7-point Likert scale to measure all the items, ranging from 1 (‘strongly disagree’) to 7 (‘strongly agree’). The initial questionnaire items were developed in English, then a blind translation/back-translation method [67] was used to ensure the accuracy of translation and expression (translating into Chinese, then back-translating into English). The entire process was reviewed and checked by two university tourism management professors.

The questionnaire consisted of six sections. A filter question ‘Are you a Macao resident?’ was used in the first section to ensure the respondents qualified. Only those responding ‘yes’ to the question were invited to complete the questionnaire. The questions in the second to the fifth sections were used to measure the four constructs. The sixth section contained questions on the background information of the respondents. Before the main survey, at the end of December 2020, a pilot test with 50 Macao residents was conducted to confirm the content. This process enabled the wording of some of the items of the questionnaire to be improved. One reversed measurement item, ‘If I need to travel in the short/medium term, I intend not to travel to cities outside the GBA’, was deleted due to its lower reliability (Cronbach’s alpha < 0.7) in the results of the pilot test. The 27 measurement items are listed in Table 1.

### 3.3. Data Collection

The 2020 statistics from the Macao Special Administrative Region government show that there are eight parishes (Nossa Senhora de Fátima, Santo António, Sé, São Lázaro, São Lourenço, São Francisco de Xavier, Nossa Senhora do Carmo, and CoTai) in Macao. Five trained research assistants systematically collected 304 questionnaires in the above parishes. The research assistants went together to one district each day. The survey was conducted from 19 to 26 February 2021 (eight days). In order to reach different groups of people of all ages, they took turns to wait in front of parks, playgrounds, dining areas, cinemas, shopping malls, and other places frequented by residents for leisure activities from 10:30 to 19:30. They identified one target out of every ten people who passed by, and told the target that the survey was for Macao residents. If the target refused to answer because he/she was not a Macao resident or for other reasons, the research assistants would choose a target from another ten people for the survey. The respondents received a small gift of US$2. It took about ten minutes for a target to fill out a questionnaire, but some respondents were older and were slower at filling out the questionnaire, taking 20 min, so the data collection process was slow. From the 304 questionnaires, 26 questionnaires were removed because they had similar answers for most items. Table 2 shows the sample profile.

## 4. Results

In current tourism research, researchers advocate the ways of using higher-order models because higher-order models can reflect more complex structures and reduce the error caused by first-order dimensions in the specific indicators [68]. In this study, ‘low-risk perception’ consists of two dimensions, and ‘perceived benefit’ consists of three dimensions. A second-order model can treat each dimension as an important part of its construction [69]. Based on this, partial least squares structural equation modeling (PLS-SEM) is used for modeling and analysis. PLS-SEM is more effective for analyzing a multi-level model [70]. Therefore, PLS-SEM was selected for analyzing the data.

### 4.1. Outer Model Analysis

The results summarized in Table 3 show the descriptive statistics for the 27 items. The minimum value of the PLS factor loading is 0.793 (>0.700). Table 4 reports that the values of Cronbach’s alpha and composite reliability (CR) for each construct are above the threshold of 0.700 [71]. This shows that the reliability is good. The value of the average variance extracted (AVE) for each construct is higher than the recommended 0.500, which, according to Hair et al. [72], shows that there is good convergent validity. Discriminant validity was examined by comparing the square root of AVE for each construct with the correlations between the pairs of latent variables [73]. Furthermore, all Heterotrait–Monotrait (HTMT) correlations in Table 5 are less than 0.900 [71], which is a satisfactory result.

### 4.2. Inner Model Analysis

The results of the PLS-SEM analysis are shown in Figure 2. Low-risk perception (LRP) of GBA travel has a significant influence on attitude (AT) towards GBA travel (β = 0.203, *p*-value < 0.001). LRP does not have any significant effect on behavioral intention (BI) (β = 0.046, *p*-value = 0.334). Perceived benefit (PB) of GBA travel has a significant influence on AT (β = 0.662, *p*-value < 0.001). PB has a significant influence on BI towards GBA travel (β = 0.362, *p*-value < 0.001). AT has a significant influence on BI (β = 0.471, *p*-value < 0.001). The R^2^ values of PB, BI, AT, and LRP are higher than 0.250. LRP has no significant effect on BI, but the other four hypotheses (H1, H3, H4, and H5) are supported. Detailed path relationships are provided in Table 6. The multicollinearity was tested by measuring the variance inflation factor (VIF). No VIF value exceeds 5, indicating that there is no multicollinearity issue [74].

The indirect effects of low-risk perception and perceived benefit show that low affective risk perception (LARP) has the greatest impact on AT, with a coefficient value of 0.118. However, low cognitive risk perception (LCRP) has a greater impact on BI (coefficient value 0.376). In the path ‘PB → AT’, the coefficient value for ‘relaxation benefit (RB) → AT’ is the largest, at 0.369, which means that RB has the greatest impact on AT, as shown in Table 7. Similarly, RB also has the greatest impact on BI. 

### 4.3. Mediating Effect on Attitude towards GBA Travel

As shown in Table 8, AT mediates both the positive impact of LRP on BI (indirect effect = 0.096, *p*-value < 0.01) and the positive impact of PB on BI (indirect effect = 0.312, *p*-value < 0.001). 

## 5. Discussion and Conclusions

### 5.1. Conclusions

The results show that, during the COVID-19 pandemic, both low-risk perception and perceived benefit have a positive influence on tourists’ attitude towards undertaking travel to the GBA. Compared with low-risk perception, perceived benefit has a greater impact on tourists’ attitude. In addition, the perceived benefit has a direct influence on behavioral intention. This is consistent with the results of previous research on the impact of perceived benefit on online shopping behavioral intention [75]. Low-risk perception does not have any direct influence on behavioral intention. The results of this study are different from those of Bae and Chang [14], who found that (high-) risk perception had a notable influence on behavioral intention. 

In the comparison of the two dimensions of low-risk perception, the impact of low affective risk perception is greater than that of low cognitive risk perception. Regarding the three dimensions of perceived benefit, it is obvious that relaxation benefit has the largest effect, followed by value benefit and convenience benefit. However, for risk perception, researchers have found that cognitive risk perception is the dominant form of risk perception [14,76,77].

### 5.2. Theoretical Contributions

This study highlights the importance of perception, perceived benefit, and attitude to the behavioral intention toward regional tourism under the environment of the pandemic. 

In the part of theoretical, researchers have tested negative attitudes towards risk. For example, in research into Koreans’ risk perception and behavioral intention to travel, (high) risk perception plays a negative role [14]. Until this research, many studies have investigated how to avoid risks during travel, but no study in the field of tourism studies has looked at the concept of low-risk perception of a place in a positive perspective. This research moves the concept of risk perception from the negative side to the positive side by introducing the concept of low-risk perception. After testing the concept of low-risk perception, other researchers can consider risk perception in two ways: high-risk perception and low-risk perception, and can therefore apply either the concept of high-risk perception or that of low-risk perception, or both, to study the phenomenon and the problems of tourists’ choice of tourism destinations.

As regards low-risk perception, this study confirms that both low affective risk perception and low cognitive risk perception exist and affect people’s attitude towards GBA travel. However, when people consider taking a trip to a certain place, if they have a high-risk perception this influences their behavioral intention [14] but if they have a low-risk perception this does not have any influence. From previous studies, we know that a high-risk perception hinders tourists’ behavioral intentions, and that people’s worries about the threats from COVID-19 are greater than their affective risk perception (such as their concern for their family). We learn from this study that low-risk perception does not promote tourist behaviors, and that people perceive travel to be low-risk from their affective risk perception of the threats of COVID-19 more than from their cognitive risk perception. Therefore, this study contributes to risk perception theory in tourism risk management research and raises a question for researchers to investigate how to promote the recovery of tourism from a low-risk perspective with positive attitudes under the pandemic.

Looking at perceived benefit, it is surprising that there has been no research testing the effect of perceived benefit on alternative tourism. According to Chen and Petrick [11], travel benefits have positive indirect effects on the frequency of travel. This study investigates the effect of travel benefits on alternative tourism. This study contributes to our understanding that perceived benefit is very important in accelerating the recovery of tourism during and after the COVID-19 pandemic. This study confirms that convenience benefit, value benefit, and relaxation benefit encourage people to undertake GBA travel. Furthermore, some destination governments offer economic benefits to lure people to travel, because they think that the economic benefit is the main benefit for people who choose to travel during and after the COVID-19 pandemic. This study corrects this thinking and empirically indicates that relaxation benefit is a major determinant for whether people travel during and after the COVID-19 pandemic. Travel has been repeatedly suppressed for long periods, so people will tend to travel for relaxation, to release the stress arising from COVID-19. This study contributes to the research on tourism recovery strategies after the COVID-19 pandemic by showing that perceived benefit, especially perceived relaxation benefit, is the main driving force to encourage people to travel during and after the COVID-19 pandemic.

To solve the issues of undertaking travel during the COVID-19 pandemic, researchers have suggested different forms of tourism such as tourism bubbles [78]. With the advent of vaccines, ‘vaccine passports’ may help the tourism recovery. However, many of the suggested forms of tourism are facing practical difficulties. For example, the Hong Kong air travel bubble with Singapore has been suspended many times. By testing Macao residents’ intention to undertake travel to the GBA, this study provides evidence to support the implementation of regional travel. Therefore, this study contributes to tourism recovery research by identifying the factors that drive people to undertake regional travel as an alternative form of tourism during the COVID-19 pandemic. It provides a reference point for researchers who are planning to study different forms of tourism during and after the COVID-19 pandemic. 

### 5.3. Managerial Recommendations

The COVID-19 pandemic has suppressed people’s travel plans, but the public willingness to travel is still very strong. From a practical point of view, during the COVID-19 pandemic, relaxation is a major motivation, and people intend to travel short distances. For this reason, cities in the GBA should create a more relaxed atmosphere and increase the promotion of tourism to attract tourists, who prefer to relieve stress and maintain mental health from within the region. As the travel distance is short, the time saved in traveling could be used by tourists to integrate themselves into the local life for in-depth leisure travel. Therefore, the governments of GBA cities should also explore the cultural connotations of these tourist destinations. As regards economic concessions, the governments should continue with their various subsidies, discounts, and other economic preferential activities.

The normalization of the prevention and control of COVID-19 could provide people with a feeling that GBA travel is low-risk. However, in order to make people have a low perception of the risk, the governments of GBA cities should regularly release transparent information and disclose in detail the current COVID-19 situation in the entire region. Secondly, all tourism enterprises should maintain good health management services, health testing, temperature measurement, disinfection and ventilation. Tourist attractions need to control the flow of visitors. These preventive practices can reduce people’s psychological worries about traveling in the GBA. The tourism companies should strengthen risk management, from a low-risk perspective, transform public health awareness and further promote tourism recovery.

### 5.4. Limitations

This research has certain limitations. First, the research setting was the GBA, and the data were collected in Macao, which is relatively safe compared with western cities. To generalize the findings, further studies in western cities are recommended. Secondly, this study only tested the effects of low-risk perception and the perceived benefit of regional travel. In future research, it would be worth exploring the antecedents of low-risk perception and perceived benefit, including social influence [79]. Furthermore, since the data were collected at a single time, a longitudinal study is suggested to examine whether the effect of perceived benefit diminishes over time. This study explores how low-risk perception and perceived benefit affect travel attitude and behavioral intention towards alternative tourism. Since no population information of GBA is available, it is unknown if the responder’s background is consistent with the characteristics of the real population under study. It is also a limitation of the study that needs to be supplemented. Future research could include personal factors such as socio-demographic characteristics and past travel experience. In short, this research provides a knowledge base, especially by looking at risk management from a positive perspective. Future research should be based on this, in order to continue to deepen the understanding of tourist decision-making.

## Figures and Tables

**Figure 1 ijerph-18-09422-f001:**
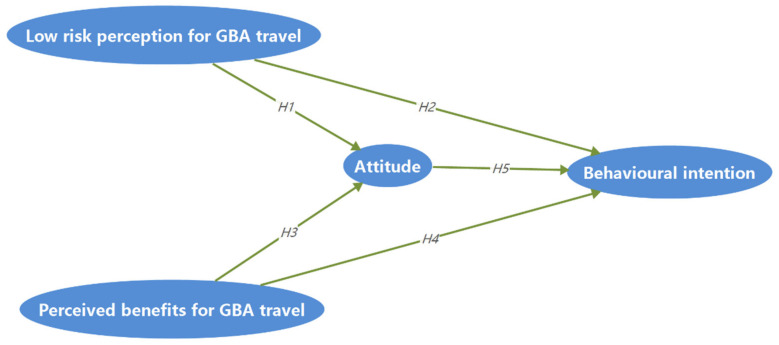
Research model.

**Figure 2 ijerph-18-09422-f002:**
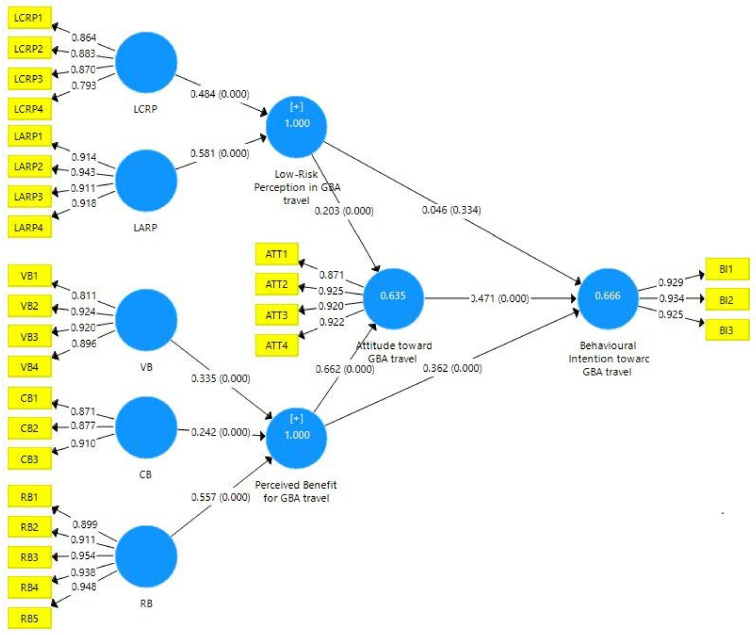
Results of PLS-SEM analysis.

**Table 1 ijerph-18-09422-t001:** Measurement scales.

	Measured Item
	Low cognitive risk perception (LCRP)
LCRP1	There is a low likelihood of acquiring COVID-19 when I travel to cities in the Greater Bay Area in general.
LCRP2	There is a low likelihood that I will acquire COVID-19 when I travel to cities in the Greater Bay Area compared to other people.
LCRP3	There is a low likelihood of acquiring COVID-19 compared to other diseases when I travel to cities in the Greater Bay Area.
LCRP4	If I travel to cities in the Greater Bay Area there is a low likelihood of dying from COVID-19.
	Low affective risk perception (LARP)
LARP1	I am not worried that I will contract COVID-19.
LARP2	I am not worried about my family members contracting COVID-19.
LARP3	I am not worried about COVID-19 occurring in my region.
LARP4	I am not worried about COVID-19 emerging as a health issue.
	Value benefit (VB)
VB1	I think the current travel to cities in the Greater Bay Area will have a large serving experience.
VB2	I think the current price of travel to cities in the Greater Bay Area will be more affordable.
VB3	I think the current reason to go to cities in the Greater Bay Area will be more reasonable prices: accommodation, food, shopping, etc.
VB4	I think the current travel to the Greater Bay Area will be value for money.
	Convenience Benefit (CB)
CB1	I think it will be more convenient to travel to cities in the Greater Bay Area for eating at present.
CB2	I think the current traffic in cities in the Greater Bay Area is easy accessibility and not congested.
CB3	I think the current travel to cities in the Greater Bay Area will have more prompt services.
	Relaxation Benefit (RB)
RB1	I think the current travel to cities in the Greater Bay Area will help renew my energy.
RB2	I think a current travel to cities in the Greater Bay Area will help get away from tedious daily life.
RB3	I think a current travel to cities in the Greater Bay Area will help relieve my current tension or stress.
RB4 RB5	I think a current travel to cities in the Greater Bay Area will help to refresh myself.
	I think a current travel to cities in the Greater Bay Area will help me relax.

**Table 2 ijerph-18-09422-t002:** Respondents’ background (*n* = 278).

		Frequency	Percent
Gender	Male	86	30.9%
	Female	192	69.1%
Age	18–29	96	34.5%
	30–41	123	44.2%
	42–53	43	15.5%
	54–65	10	3.6%
	Over 65	6	2.2%
Education	Junior high school or below	96	34.5%
(Completed)	High school	123	44.2%
	Associate degree/diploma	43	15.5%
	Bachelor degree	10	3.6%
	Master or above	6	2.2%
Monthly income	Less than 1200	72	25.9%
(USD)	1200–2400	71	25.5%
	2401–3600	56	20.1%
	3601–4800	46	16.5%
	4801–6000	21	7.6%
	6001 or over	12	4.3%
Number of trips to cities in the Greater Bay Area (In the past year, excluding Zhuhai)	0	102	36.7%
	1	54	19.4%
	2	48	17.3%
	3	19	6.8%
	4	4	1.4%
	5 or above	51	18.3%

**Table 3 ijerph-18-09422-t003:** Descriptive statistics and factor loadings.

	Mean	S.D.	Kurtosis	Skewness	Loadings
LCRP1	4.363	1.532	−0.337	−0.355	0.864
LCRP2	4.119	1.622	−0.796	−0.147	0.883
LCRP3	4.219	1.600	−0.825	−0.193	0.870
LCRP4	4.230	1.693	−0.903	−0.097	0.793
LARP1	4.255	1.654	−0.982	−0.202	0.914
LARP2	4.162	1.700	−1.027	−0.127	0.943
LARP3	3.924	1.723	−1.034	0.007	0.911
LARP4	3.982	1.674	−1.082	−0.055	0.918
VB1	4.572	1.333	0.204	−0.361	0.811
VB2	4.669	1.338	0.021	−0.520	0.924
VB3	4.727	1.310	0.178	−0.516	0.920
VB4	4.629	1.293	0.509	−0.559	0.896
CB1	4.698	1.425	−0.402	−0.400	0.871
CB2	4.601	1.337	−0.267	−0.376	0.877
CB3	4.583	1.277	−0.157	−0.243	0.910
RB1	4.608	1.389	0.155	−0.512	0.899
RB2	4.658	1.336	0.290	−0.619	0.911
RB3	4.784	1.345	0.513	−0.724	0.954
RB4	4.770	1.345	0.616	−0.768	0.938
RB5	4.863	1.321	0.569	−0.726	0.948
AT1	4.022	1.486	−0.236	−0.183	0.871
AT2	4.360	1.391	0.074	−0.405	0.925
AT3	4.727	1.327	0.808	−0.807	0.920
AT4	4.784	1.337	0.555	−0.734	0.922
BI1	4.982	1.410	0.743	−0.896	0.929
BI2	4.647	1.514	−0.225	−0.462	0.934
BI3	4.975	1.443	0.750	−0.996	0.925

**Table 4 ijerph-18-09422-t004:** Reliability, construct validity, and correlation.

	Cronbach’s Alpha	CR	AVE	Fornell-Larcker Criterion						
				AT	BI	LARP	LCRP	CB	VB	RB
Attitude toward GBA travel (AT)	0.930	0.950	0.827	*0.910*						
Behavioral intention toward GBA travel (BI)	0.921	0.950	0.864	0.781	*0.929*					
Low affective risk perception (LARP)	0.941	0.957	0.849	0.557	0.501	*0.921*				
Low cognitive risk perception (LCRP)	0.875	0.914	0.728	0.539	0.493	0.761	*0.853*			
Convenience benefit (CB)	0.863	0.917	0.785	0.573	0.554	0.441	0.453	*0.886*		
Value benefit (VB)	0.911	0.938	0.790	0.597	0.560	0.413	0.466	0.742	*0.889*	
Relaxation benefit (RB)	0.961	0.970	0.865	0.790	0.779	0.505	0.522	0.613	0.626	*0.930*

**Table 5 ijerph-18-09422-t005:** Heterotrait–monotrait ratio.

	Heterotrait–Monotrait Ratio					
	AT	BI	LARP	LCRP	CB	VB
Attitude toward GBA travel (AT)						
Behavioral intention toward GBA travel (BI)	0.840					
Low affective risk perception (LARP)	0.595	0.538				
Low cognitive risk perception (LCRP)	0.595	0.549	0.836			
Convenience benefit (CB)	0.637	0.620	0.489	0.519		
Value benefit (VB)	0.648	0.612	0.447	0.522	0.834	
Relaxation benefit (RB)	0.833	0.827	0.531	0.569	0.670	0.669

Remark: AVE—average variance extracted; CR—construct reliability, *Italic* front-square-root of AVE.

**Table 6 ijerph-18-09422-t006:** Results of hypotheses testing.

	Coefficient	T-Statistics	f-Square	VIF	Test Results
H1 LRP → AT	0.203	4.058	0.076	1.495	Supported
H2 LRP → BI	0.046	0.945	0.004	1.609	NOT Supported
H3 PB → AT	0.662	13.522	0.804	1.495	Supported
H4 PB → BI	0.362	4.743	0.146	2.698	Supported
H5 AT → BI	0.471	6.087	0.243	2.741	Supported

Remark: LRP—low-risk perception, AT—attitude toward GBA travel, BI—behavioral intention toward GBA travel, PB—perceived benefit.

**Table 7 ijerph-18-09422-t007:** Indirect effects of perceived benefit and low-risk perception.

Indirect Effect	Coefficient	T-Statistics
LCRP→AT	0.099	4.029
LARP→AT	0.118	4.071
LCRP→BI	0.069	2.557
LARP→BI	0.083	2.586
VB→AT	0.226	12.142
CB→AT	0.160	11.526
RB→AT	0.369	14.480
VB→BI	0.222	11.623
CB→BI	0.163	11.920
RB→BI	0.376	14.997

**Table 8 ijerph-18-09422-t008:** The mediation effect of attitude toward GBA travel.

	Effect	*p* Value	2.5%	97.5%	Mediation
LRP→AT→BI	0.096	0.001	0.046	0.154	Full mediation
PB→AT→BI	0.312	0.000	0.198	0.434	Partial mediation

Remark: PB—perceived benefit, AT—attitude toward GBA travel, BI—behavioral intention toward GBA travel, LRP—low-risk perception.

## Data Availability

Data from this study are available from the corresponding author on reasonable request.

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
