# Peer review of "Regional Travel as an Alternative Form of Tourism during the COVID-19 Pandemic: Impacts of a Low-Risk Perception and Perceived Benefits"

_ijerph, 2021, doi:10.3390/ijerph18179422_

Round 1

Reviewer 1 Report

The article is interesting and it is very well structured as a scientific study.

The hypotheses are well formulated and the questions allow the collection of data for the authors' reasoning.

The doubts I had while reading the article turned out to be the same ones the authors have regarding 5.4 Limitations, hence I have nothing to add.

Author Response

Thank you so much for your kind evaluations! We have received the comments and will continue to improve ourselves. Thanks again for your recognition and support!

Reviewer 2 Report

The work is extremely vague in its content.

Section 2, Literature Review, says practically nothing and is full of the obvious. Section 2.1 would remain complete, and 2.2.1 only exposes and proposes obvious hypotheses. Special deserves the Research Model, very simple, and obvious.

Some issues should be treated with greater attention. This is the case of risk in tourism, on this issue, there is abundant research, also conceptual on the subject, which concludes the inherent nature of risk in services and tourism. The authors even suggest the non-existence of research on Low Riks: it is clear that the multiple existing empirical works speak of risk levels, consequently, these levels can range from very high to very low.

It is unknown if the responder's background is consistent with the characteristics of the real population under study.

The problem of this work already begins in the title itself: it is not possible to speak of ALTERNATIVE in tourism during the Covid, simply because international tourism does not exist.

Finally, he does not see the fit of the content in a magazine whose fields are the environment and public health.

Author Response

Point 1: Section 2, Literature Review, says practically nothing and is full of the obvious. Section 2.1 would remain complete, and 2.2.1 only exposes and proposes obvious hypotheses. Special deserves the Research Model, very simple, and obvious.

Response 1:

The literature has been updatedusing risk perception (Pages 3-4)

Page 3:

Among the small-scale community regional control of the tourism industry, community security affects public risk perception and further affect health tourism intention.

Page 4:

Health risks in tourism are associated with potential health hazards during travel to a place. Since previous studies in tourism risk management were of little help in researching travel during the COVID-19 pandemic, many researchers have recently studied the risks of COVID-19 in undertaking normal travel. Perhaps low-risk can be used as a positive factor in the public health field to alternative tourism in the post-epidemic era.

Point 2: Some issues should be treated with greater attention. This is the case of risk in tourism, on this issue, there is abundant research, also conceptual on the subject, which concludes the inherent nature of risk in services and tourism. The authors even suggest the non-existence of research on Low Riks: it is clear that the multiple existing empirical works speak of risk levels, consequently, these levels can range from very high to very low.

Response 2:

We agree with you that in many public safety incidents, there have been many studies about risk, completely, but it is undeniable that we have changed our perspective and selected relatively low-risk regional tourism from the environment where travel risks still exist. On the other world, negative factors are superimposed into positive factors, and we are pioneers when we put low-risk as a positive factor in the public health domain to alternative tourism in the post-epidemic era. Revisions have been made to point out hypotheses in the paragraph. (Pages 2-4; Page 13)

Page 2:

Although risk perception and risk management researches have been widely studied, there has been no study of the concept of a low-risk perception of an alternative type of travel in positive perspective.

Page 4:

Since previous studies in tourism risk management were of little help in researching travel during the COVID-19 pandemic, many researchers have recently studied the risks of COVID-19 in undertaking normal travel. Perhaps low-risk can be used as a positive factor in the public health field to alternative tourism in the post-epidemic era.

Page12:

In the part of theoretical, earlier researchers have tested negative attitudes towards risk. For example, in research into Koreans’ risk perception and behavioural intention to travel, (high) risk perception plays a negative role. Until this research, many researches study how to avoid risks during travel, but no study in the field of tourism studies has looked at the concept of low-risk perception from low-risk in an unsafe environment, double negation means affirmation as a positive perspective. 

Point 3: It is unknown if the responder's background is consistent with the characteristics of the real population under study.

Response 3:

For unknowing the consistent of responder's background with the characteristics of the real population under study, it is a limitation of the study. (Page 14)

Page 14:

Since no population information of GBA is available, it is unknown if the responder's background is consistent with the characteristics of the real population under study. It is also a limitation of the study that needs to be supplemented.

Point 4: The problem of this work already begins in the title itself: it is not possible to speak of ALTERNATIVE in tourism during the Covid, simply because international tourism does not exist.

Response 4:

Regarding the wording of ‘alternative’, we found the relevant usage in the paper of  ‘Contextualizing the complexities of managing alternative tourism at the community-level: A case study of a nordic eco-village.’ (Solene, P.; Dimitri, I. Contextualizing the complexities of managing alternative tourism at the community-level: A case study of a nordic eco-village. Tour Manag. 2017, 60. https://doi.org/10.1016/j.tourman.2016.12.015.) (c) (Page 1)

Point 5: Finally, he does not see the fit of the content in a magazine whose fields are the environment and public health. 

Response 5:

Previous research has shown that anxiety is a crucial determinant of behaviour, when people become more anxious about the virus, they tend to maintain a high level of personal hygiene, more social distancing and are more likely to get vaccinated when available. In addition, they may overstock necessities, conduct unnecessary medical tests, or misinterpret their minor symptoms as signs of serious infection. Anxiety is a mental state of tension and worry about the future and high anxiety might act as a signal to avoid taking risks. The outbreak of COVID-19 does not only make people anxious, but it also scares people, the fear of the pandemic recently would affect people’s travel behaviour. COVID-19 does not only affect tourism but also the global economy. Researchers have studied the general effect of COVID-19 on tourism. On the whole, it is affecting public physical and mental health. (Page 2-3)

Page2:

The pandemic has brought the world's travel traffic to a standstill, and many countries have banned foreigners from entering the country and closed their foreign borders. The degree of rigour (or looseness) of COVID-19 prevention and control enables us to set up three categories: China’s national strictly precise prevention and control model; Europe’s flexible prevention and control model; and America’s free and loose prevention and control model. Restrictions on freedom of movement, closure of borders, and people's fear of infection have completely caused people’s anxiety and greatly impacted the tourism industry. The Asmundson’s research shows that anxiety, a state of tension and worry, mainly affects people's behavioral intentions. In this epidemic, the tension and anxiety caused by policies such as restricting free movement have even made people feel fear. On the whole, it is affecting public physical and mental health.

Traveling can relieve psychological pressure and regulate anxiety. During the recovery of the tourism market, the attraction radius of tourist destinations has shrunk, the proportion of local tourists has increased, and long-distance tourists have dropped significantly.

Reviewer 3 Report

Dear Authors,

thanks for allowing me to review your manuscript on domestic travels during the pandemic. The manscript addresses the issue that during the pandemic people could not travel abroad, therefore choose domestic destinations. The implications of this study are interesting, but I would not underline so many times that "this is the first study to do this or that" because the topic of Covid-19 has "exploded" research in every area. If you claim that your study is the first to address domestic tourism, it might be that you did not find the other studies exploring the same.

Moreover, past studies on the effects and implications of SARS (e.g. McKercher & Pine, 2006) could be interesting to your study as well. In addition, travel craving (see Mitev, A. and Irimiás, A. 2020. “Travel Craving.” Annals of Tourism Research https://doi.org/10.1016/j.annals.2020.103111) could be one of the reasons why tourist "choose the less rotten apple".

In the conclusion, insightful theoretical contributions and useful practical implications to the tourism academics and industry should be presented.

Author Response

Point 1:  I would not underline so many times that "this is the first study to do this or that" because the topic of Covid-19 has "exploded" research in every area. If you claim that your study is the first to address domestic tourism, it might be that you did not find the other studies exploring the same.

Response 1:

The introduction was revised to explain the pioneer of the study. (Pages 2-4; Page 13)

The 4th paragraph of the introduction brought out the question of ‘this is the first study to do this or that’.

The 2th paragraph of the literature review about the risk perception, the paragraph of the introduction brought out ‘the linkage between international students’ place attachment and WOM generation.’ It is a human-place relationship.

Page 2:

Although risk perception and risk management research has been widely studied, there has been no study of the concept of a low-risk perception of an alternative type of travel in a positive perspective.

Page 4:

Since previous studies in tourism risk management were of little help in researching travel during the COVID-19 pandemic, many researchers have recently studied the risks of COVID-19 in undertaking normal travel. Perhaps low-risk can be used as a positive factor in the public health field to alternative tourism in the post-epidemic era.

Page13:

In the part of theoretical, earlier researchers have tested negative attitudes towards risk. For example, in research into Koreans’ risk perception and behavioural intention to travel, (high) risk perception plays a negative role. Until this research, many previous studies investigate how to avoid risks during travel, but no study in the field of tourism studies has looked at the concept of low-risk perception of a place in a positive perspective.

We agree with you that in many public safety incidents, there have been many studies about risk, completely, but it is undeniable that we have changed our perspective and selected relatively low-risk regional tourism from the environment where travel risks still exist. On the other world, negative factors are superimposed into positive factors, and we are pioneers when we put low-risk as a positive factor in the public health domain to alternative tourism in the post-epidemic era.

Point 2: Moreover, past studies on the effects and implications of SARS (e.g. McKercher & Pine, 2006) could be interesting to your study as well. In addition, travel craving (see Mitev, A. and Irimiás, A. 2020. “Travel Craving.” Annals of Tourism Research https://doi.org/10.1016/j.annals.2020.103111) could be one of the reasons why tourist "choose the less rotten apple".

Response 2:

The concept of ‘travel craving’ has been added to the 4th paragraph of the introduction. (Page 2)

Pages 2:

During the COVID-19, the travel craving, which also enables people to choose a good physical and mental tourism activities in low-risk areas. These cognitive emotional activities have also played a certain promotion of the recovery of tourism.

Point 3: In the conclusion, insightful theoretical contributions and useful practical implications to the tourism academics and industry should be presented.

Response 3:

The contributions for theoretical and managerial were revised. (Pages 12-14)

The theoretical contribution is written in three directions: the content that the ancestors have studied, the content of this research, and the perspectives that future researchers can start with. And starting from the three levels of the market, government, and enterprises, make recommendations related to the tourism industry.

Pages 12:

This study highlights the importance of perception, perceived benefit and attitude to the behavioural intention toward regional tourism under the environment of the epidemic.

In the part of theoretical, researchers have tested negative attitudes towards risk. For example, in research into Koreans’ risk perception and behavioural intention to travel, (high) risk perception plays a negative role. Until this research, many previous studies have investigated how to avoid risks during travel, but no study in the field of tourism studies has looked at the concept of low-risk perception of a place in a positive perspective...this study contributes to risk perception theory in tourism risk management research and raises a question for researchers to investigate: how to promote the recovery of tourism from a low-risk perspective with positive attitude under the pandemic... It provides a reference point for researchers who are planning to study different forms of tourism during and after the COVID-19 pandemic.

Page 14:

From a practical point of view, during the COVID-19 pandemic, relaxation is a major motivation, and people intend to travel short distances...cities in the GBA should create a more relaxed atmosphere and increase the promotion of tourism to attract tourists, who prefer to relieve stress and maintain mental health from within the region...the governments of GBA cities should also explore the cultural connotations of these tourist destinations...the governments should continue with their various subsidies, discounts, and other economic preferential activities...the governments of GBA cities should regularly release transparent information and disclose in detail the current COVID-19 situation in the entire region... Tourist attractions need to control the flow of visitors...And the tourism companies should strengthen risk management, from low-risk perspectives, transform public health awareness and further promote tourism recovery.
